# A large-scale genome-wide association meta-analysis for nevus count provides direct insights into the genetics of melanoma

G. J. M. Shanika R. Jayasinghe [1,2] ✉, Gu Zhu[3], Nirmala Pandeya [4,5],
Catherine M. Olsen [4,6], Nicholas G. Martin[3], Penelope A. Lind [2,7,8],
Sarah E. Medland[2,7,9], Scott D. Gordon[3], Santiago Diaz-Torres[2,10],
Gareth Lingham [11,12,13], Samantha S. Y. Lee [11], Tamar Nijsten[14],
Manfred Kayser[15], Luba M. Pardo[14], Grant W. Montgomery [16],
Nicholas K. Hayward[17], Jane M. Palmer[17], David J. Hunter[18], Jiali Han[19,20],
Alex W. Hewitt [21,22], Mario Falchi [23], D. Timothy Bishop [24],
Kevin M. Brown [25], Veronique Bataille[23,26], David A. Mackey [11,21,22],
Mark M. Iles [24,27], David C. Whiteman[4], David L. Duffy[3,6], Stuart MacGregor[2,10] &
Matthew H. Law [1,2,8] ✉

A greater understanding of the biology of nevi will provide insights into the etiology of melanoma. Our large-scale meta-analysis of 14 nevus genome-wide association studies (GWAS) includes 85,965 individuals of European ancestry. We identify 29 nevus-associated loci ($p < 5 \times 10^{-8}$), of which 24 have not been previously reported in a GWAS conducted for nevus count alone. We further identify 255 candidate genes for nevus loci, including *SIKE1* which is involved in immune response regulation. This is of interest because immune response regulation influences the formation of nevi and melanoma susceptibility. Gene-set enrichment analyses prioritise immune response-related pathways and cancers that do not have a pigmentation component (e.g. breast, prostate, and glioma). This suggests that the biology underlying nevus count captures risk pathways beyond pigmentation that are relevant to melanoma. In sex-specific analyses, we observe higher total-body nevus count in females than in males, however the genetic architecture is largely shared (genetic correlation = 0.863, 95% CI = 0.453 – 1.273), indicating the difference may be influenced by environmental and behavioural factors rather than genetics. A nevus polygenic risk score explains 5% of the variance in nevus count, indicating its potential to enhance melanoma risk prediction.

Melanocytic nevi are small, pigmented, benign skin tumors composed of melanocytes. Melanoma is the deadliest form of skin cancer and arises from the malignant transformation of melanocytes. Melanoma can develop from a pre-existing nevus or appear as a new lesion on the skin[1,2]. A high melanocytic nevus count is the strongest single risk factor for melanoma[3]. Therefore, understanding the biology of nevi may also provide insights into the etiology of melanoma.

Nevus count is a highly heritable trait with heritability estimates from twins ranging from 60 to 70%[4]. Previous genome-wide association studies (GWAS) for nevus count (n > 52,000) have identified five

**Fig. 1 | Manhattan plot of the GWAS of nevus count.** Each dot represents a variant, plotted by chromosome location (x-axis) and the −log$_{10}$ (P) of association (two-sided Z-test) (y-axis). The red dashed line shows the genome-wide significance threshold of multiple testing corrected $P$ ($5 \times 10^{-8}$). Novel loci are represented in the plot in red, whereas loci reported in previous nevus GWASs, nevus/melanoma or nevus/pigmentation combined GWASs are presented in blue.

significant loci[5]. These loci were also associated with melanoma risk[5], implying both traits have a shared genetic risk component. Although combined analyses of both melanoma and nevus count have identified additional loci that may be pleiotropically associated with both traits[5,6], given the high heritability of nevus count, many causal genetic variants remain to be discovered.

Nevus counts show sex-specific patterns by body site, with males exhibiting higher counts on the head, neck, and trunk, and females often showing higher counts on the limbs, particularly the lower limbs[7]. These patterns mirror sex-specific differences observed in the body site distribution of melanoma[7], and both behavioral and genetic factors may contribute to them. Therefore, evaluating potential phenotypic and genetic differences in nevus count between the sexes provides biological insight into the determinants of nevus formation and helps clarify whether sex-specific pathways contribute to downstream melanoma susceptibility.

In addition to providing insights into cutaneous melanoma, nevus count genetics can improve our understanding of rarer forms of melanoma. Uveal melanoma is a rare but deadly form of melanoma originating from ocular melanocytes. Atypical nevi and iris nevi are risk factors for uveal melanoma[8–10], suggesting that the findings from a cutaneous nevus GWAS can be applied to risk factors for uveal melanoma.

To identify further genetic variants associated with nevus count, we meta-analyzed 14 GWASs of nevus count, comprising over 85,000 individuals, including two previously unpublished datasets. We also report on the application of post-GWAS functional analyses to identify those genes most strongly involved in nevus biology and to identify the pathways underlying these genes. We also conducted sex-specific GWAS meta-analyses of nevus count that vary by sex and explored the degree of causal association between melanoma and nevus count through Mendelian randomization analysis. Finally, we generated polygenic scores for nevus count and tested their ability to predict nevus count in adults and to predict iris nevus count.

## Results

### Nevus GWAS meta-analysis
We initially estimated the degree to which genotyped SNPs could capture variation in nevus count within the QSkin I cohort ($n = 15,346$). The SNP-based heritability of nevus count estimated using GCTA-GREML was 0.22 (SE = 0.03).

We then performed a large-scale GWAS meta-analysis of nevus count with a total sample size of 85,965 participants. The meta-analysis included variants on both autosomes and the X chromosome (see "Methods"); however, no genome-wide significant signals were observed on the X chromosome (Fig. 1). Therefore, subsequent analyses focused exclusively on autosomal findings. The genomic inflation factor $\lambda$ and LDSC intercept for the GWAS meta-analysis suggested polygenic inheritance ($\lambda = 1.068$, intercept = 1.014, ratio = 0.119). In the nevus meta-analysis, 41 independent genome-wide significant SNPs were assigned to 29 genomic loci (Fig. 1). Of these 29 loci, 24 have not previously been reported in a GWAS conducted solely for nevus count (Table 1). Nineteen of these 24 loci have been previously identified as associated with both nevus count and melanoma in joint analyses[5,6]. Among the remaining five loci after 19/24, the regions near *MAP3K1* on chromosome 5 and around *TPCN2* on chromosome 11 have also been associated with melanoma and hair color[6]. LDSC regression analysis estimated the proportion of the variance in nevus count captured by the GWAS SNPs in the current GWAS meta-analysis at 0.06 (SE = 0.01); while this is expected to be an underestimate relative to GCTA-GREML as it uses fewer SNPs and summary data, this estimate of variance is two times higher than that of the previous nevus GWAS meta-analysis[5].

### Pleiotropy of nevus loci
To determine whether the significant nevus loci were pleiotropic with melanoma, a pairwise GWAS was conducted by integrating our results with a previously published melanoma GWAS[6]. This approach assigns a posterior probability of association (PPA), which sums to 1, to four models; models 1 and 2 suggest the gene region is only associated with trait 1 or 2, respectively; model 3 suggests pleiotropy, and model 4 tests an independent association[11] (see "Methods"). Twenty-eight of 29 nevus loci exhibited strong evidence of pleiotropic regions for both melanoma and nevus count (model 3 PPA > 0.95) (Supplementary Data 1). Locus, near *TPCN2* on chromosome 11, had the highest PPA for model 4 (independence; PPA 0.76), but as no model reached the threshold (>0.95), we are unable to reach strong conclusions about the role of this locus.

### Functional mapping
We subsequently used FUMA (Functional Mapping and Annotation of GWASs) to identify potential functional links between nevus-associated variants and candidate genes at genome-wide significant loci ("Methods"). Using three mapping approaches (positional, eQTL,

**Table 1 | Top significant SNPs at each locus identified by the nevus GWAS meta-analysis**

| rsID | chr | Pos | EA/NEA | Freq | P | Zscore | Genes |
|---|---|---|---|---|---|---|---|
| rs7537232* | 1 | 115284056 | T/G | 0.26 | $6.80 \times 10^{-10}$ | 6.17 | BCAS2, SIKE1 |
| rs76798800 | 1 | 154994978 | T/G | 0.247 | $7.81 \times 10^{-10}$ | 6.15 | PBXIP1, ADAM15 |
| rs6546237 | 2 | 25749376 | T/C | 0.782 | $3.67 \times 10^{09}$ | −5.90 | DTNB |
| rs1800440 | 2 | 38298139 | T/C | 0.824 | $1.72 \times 10^{-9}$ | 6.02 | CYP1B1 |
| rs11677464 | 2 | 240065356 | A/G | 0.904 | $4.94 \times 10^{-9}$ | −5.85 | HDAC4 |
| rs113679968 | 4 | 37439840 | A/G | 0.147 | $4.68 \times 10^{-9}$ | −5.86 | C4orf19 |
| rs252899* | 5 | 56188569 | C/G | 0.81 | $2.37 \times 10^{-8}$ | −5.58 | MAP3K1 |
| rs251468 | 5 | 149194485 | T/C | 0.28 | $2.17 \times 10^{-9}$ | −5.99 | PPARGC1B |
| rs1808363* | 5 | 159888522 | A/G | 0.525 | $3.22 \times 10^{-9}$ | −5.92 | PTTG1 |
| rs12203592 | 6 | 396321 | T/C | 0.131 | $2.02 \times 10^{-20}$ | −9.26 | IRF4 |
| rs10949304 | 6 | 15503696 | C/G | 0.451 | $9.70 \times 10^{-13}$ | 7.14 | JARID2, DTNBP1 |
| rs117132860 | 7 | 17134708 | A/G | 0.019 | $7.88 \times 10^{-10}$ | 6.15 | AHR |
| rs4728211 | 7 | 130876086 | T/C | 0.13 | $1.06 \times 10^{-10}$ | 6.46 | MKLN1 |
| rs2120470 | 8 | 72863438 | T/G | 0.366 | $1.26 \times 10^{-11}$ | −6.77 | MSC |
| rs16904189 | 8 | 131025191 | T/C | 0.959 | $4.01 \times 10^{-12}$ | 6.94 | GSDMC |
| rs600951 | 9 | 224742 | A/G | 0.463 | $4.74 \times 10^{-12}$ | 6.91 | DOCK8 |
| rs869329 | 9 | 21804693 | A/G | 0.477 | $3.2 \times 10^{-72}$ | 17.97 | MTAP, CDKN2A |
| rs2818292 | 9 | 109011615 | A/C | 0.738 | $1.63 \times 10^{-11}$ | −6.74 | TMEM38B |
| rs10816599 | 9 | 110716721 | C/G | 0.332 | $2.44 \times 10^{-16}$ | −8.20 | KLF4 |
| rs1455118 | 11 | 16221683 | T/C | 0.516 | $7.40 \times 10^{-10}$ | 6.16 | SOX6 |
| rs11228537* | 11 | 68927358 | A/G | 0.886 | $2.31 \times 10^{-8}$ | 5.59 | TPCN2 |
| rs1640875 | 12 | 13069524 | A/T | 0.563 | $3.62 \times 10^{-15}$ | −7.87 | GPRC5D |
| rs1586869* | 12 | 88980649 | A/G | 0.827 | $3.5 \times 10^{-9}$ | 5.91 | KITLG |
| rs10873172 | 14 | 64390030 | C/G | 0.71 | $1.63 \times 10^{-13}$ | −7.38 | SYNE2 |
| rs150962800 | 15 | 33260973 | T/C | 0.017 | $1.85 \times 10^{-18}$ | 8.77 | FMN1 |
| rs12902005* | 15 | 86091174 | A/G | 0.137 | $2.90 \times 10^{-9}$ | 5.94 | AKAP13, KLHL25 |
| rs1529981 | 16 | 55323241 | T/G | 0.237 | $3.4 \times 10^{-8}$ | 5.52 | MMP2 |
| rs1805007 | 16 | 89986117 | T/C | 0.063 | $2.40 \times 10^{-12}$ | −7.01 | MC1R |
| rs133018 | 22 | 38572761 | A/G | 0.481 | $5.36 \times 10^{-36}$ | −12.53 | PLA2G6 |

*chr* chromosome, *pos* position (hg37), *EA* effect allele, *NEA* non-effect.

*P* values are from a two-sided Z-test. All loci are significant at multiple testing corrected $P < 5 \times 10^{-8}$. Information in the Gene column prioritizes the functional target gene if known previously; if not known, it reports gene(s) prioritized by all three mapping approaches or finally the nearest protein-coding gene. *Novel loci not reported in previous nevus GWASs, nevus/melanoma or nevus/pigmentation combined GWASs.

and chromatin interaction mapping) 248 genes were prioritized across the nevus count loci (Supplementary Data 2). These prioritized genes included previously reported nevus (e.g., *KITLG, MTAP/CDKN2A,* and *IRF4*) or melanoma-related genes (e.g., *MC1R, TMEM38B,* and *DCST2*)[5,6,12]. Of 248 genes, 26 overlapped in all three mapping approaches, and those genes mapped on 14 of 29 nevus-associated genomic loci (Supplementary Data 3). Interestingly, of these 26 genes, we identified candidate genes for 2/5 novel genomic regions that have not previously been reported in a nevus GWAS, nevus/melanoma combined analysis or a nevus/pigmentation combined analysis, marked by rs7537232 (*BCAS2* and *SIKE1*) and rs12902005 (*AKAP13* and *KLHL25*) as indicated in Table 1. In addition, *PTTG1* was identified as the candidate gene for the remaining nevus-associated novel locus on chromosome 5 through the positional and eQTL mapping.

**Gene-based analyses**

MAGMA gene-based analysis evaluates candidate genes for association with a trait by combining the local association test statistics for SNPs within a gene region. This approach can identify additional significant genetic loci that may not reach genome-wide significance in single SNP tests. MAGMA identified 33 genome-wide significant genes ($P < 2.64 \times 10^{-6}$) at 18 genomic loci; three were in addition to those found in our GWAS meta-analysis (Fig. 2 and Supplementary Data 4). Of these 33 genes, 28 overlapped with those prioritized by functional mapping approaches and included well-known nevus-, melanoma-, or

pigmentation-associated genes, providing further evidence of their shared genetics. The three additional loci identified by MAGMA included a novel locus not previously identified for nevus count or melanoma on chromosome 22 (*C22orf26*), a locus previously found in nevus/melanoma combined analysis on chromosome 10 (*FAM208B*)[5], and a locus previously reported for melanoma on chromosome 3 (*MYNN*)[12]. Interestingly, through MAGMA gene analysis, we found *NRAS* as a significant candidate gene for nevi at the nevus risk locus marked by rs7537232.

**Transcriptome-wide association study (TWAS)**

To expand on our gene-mapping work, we carried out TWAS to nominate plausible functional genes linked to nevus count by testing for significant correlation with gene expression. We analyzed gene expression data from three skin-related tissue-based cis-eQTL datasets (sun-exposed, not-sun-exposed, and fibroblasts) available in GTEx (see "Methods"), melanocyte-specific eQTL data[6], and an additional well-powered but potentially less tissue-relevant whole-blood dataset from the eQTLGen consortium. After applying multiple-testing corrections followed by colocalisation analysis, the five TWAS analyses identified a range of promising candidate genes. Across all TWAS analyses, candidate genes were identified at 12 genome-wide significant nevus loci and two additional loci on chromosome 10 (*LRMDA* and *GDI2*) (Supplementary Data 5.1 and 5.2). *LRMDA* was significant in non-sun-exposed skin tissue and showed increased predicted gene expression

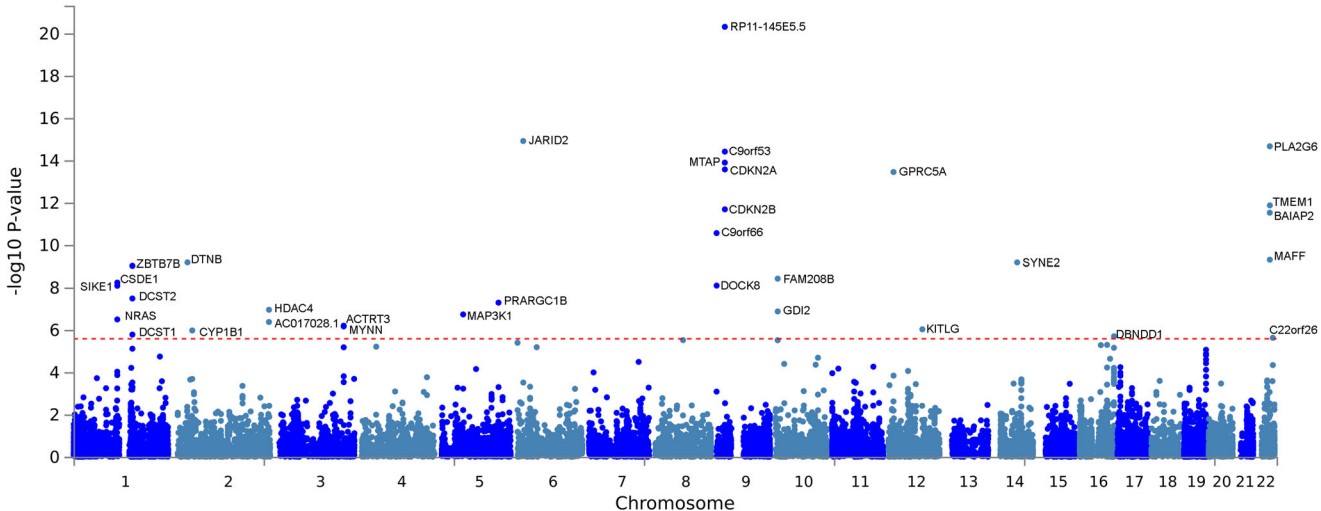

**Fig. 2 | MAGMA gene-based analysis identified 33 genes associated with nevus count.** Each dot represents a gene, the *x*-axis shows the chromosome location of each gene, and the *y*-axis shows $-\log_{10}(P)$ (two-sided; from the Z-statistic). The red dashed line shows the genome-wide significance threshold of multiple testing corrected $P$ ($0.05/19,055 = 2.64 \times 10^{-6}$).

associated with higher nevus count (Z-score = 5.061). *GDI2* was significant across sun-exposed skin, fibroblasts, and whole blood tissues with uniformly negative Z-scores (Z-score = −5.157, −5.505, and −4.950, respectively), indicating a consistent inverse association across tissues. Further, *GDI2* was replicated in MAGMA gene-based analysis (Supplementary Data 4). The consistency of signals across tissues, together with its replication across methods, provides strong evidence for implicating *GDI2* in nevus count.

### Pathway analyses

In the MAGMA gene-set analysis, three gene sets involved in the regulation of the microtubule cytoskeleton during radial glia-guided migration in the cerebral cortex, development of astrocytoma in patients with Neurofibromatosis Type 1 (NF1) and regulation of the catalytic activity (i.e., any process that modulates the activity of enzyme), were found to be significant after Bonferroni correction (Supplementary Data 6).

In addition to using the MAGMA data, we conducted gene-set enrichment tests using the genes prioritized by any of the three gene mapping approaches in FUMA. Notably, prioritized genes were significantly enriched in gene sets related to cancers (e.g., breast, prostate, and glioma cancer) that do not have a pigmentation component. As expected, gene-sets previously reported for nevus, melanoma, and pigmentation were also enriched (Supplementary Data 7). Fifteen genes (including *NRAS, THBS3, IFNA16, CCND1,* and *KITLG*) were enriched in the critical cell survival, growth, and apoptosis P13K/AKT signaling pathway (adjusted $P = 1.68 \times 10^{-5}$). In addition, we found significant enrichments in several immune response-related pathways. The top two pathways were the type 1 interferon receptor binding pathway and RIG-I-like (Retinoic acid-inducible gene I) receptor signaling pathway (Supplementary Data 7). Further, 28 prioritized genes were significantly overrepresented in the gene set involved in response to the organic cyclic compounds. Of transcriptional factor target gene sets, four gene sets showed significant enrichment signals, of which the most significant was the target gene set regulated by transcription factor PRDM4 (PR domain zinc finger protein 4), which is involved in cell proliferation and tumourigenesis[13,14].

### Sex-specific meta-analysis GWAS for nevus count

We found differences between males and females for nevus count in QSkin I, which consists of 6769 males and 8577 females. To test the difference, we fitted an ordinal logistic regression model for nevus count on a 4-point scale (none, few, moderate, more) and sex after adjusting for age, and ancestry (first 10 PCs). Sex was associated with the nevus count, and females had significantly higher odds of being in a higher nevus count category compared with males (OR = 1.294, 95% CI = 1.216−1.377).

Then we performed a sex-stratified GWAS meta-analysis (*N* for men = 11,963; *N* for females = 21,766) integrating QSkin I, QSkin II, and AGDS. Sex-specific GWAS identified nine genome-wide significant loci in females and four in males, with three loci overlapping between the GWASs (Supplementary Fig. 20). We identified two loci (rs10873172[*SYNE2*] on chromosome 14 and rs35096708[*FANCA*] on chromosome 16) with significantly different effect sizes by sex ($P < 0.005$, Bonferroni-corrected $P$ value for 10 tests) as indicated in Supplementary Data 8. However, genetic correlation between males and females for nevus count calculated from LDSC regression was not significantly different from 1 ($r_g = 0.863$, 95% CI = 0.453−1.273, $P = 3.86 \times 10^{-5}$).

### Causal association with melanoma

Mendelian randomization (MR) showed that genetically predicted higher nevus count was significantly associated with a higher risk of melanoma (Inverse variance weighted (IVW) OR = 4.212, 95% CI = 3.469−5.114). Sensitivity analysis yielded consistent effect estimates with overlapping confidence intervals, reinforcing the robustness of the association (Supplementary Table 1 and Supplementary Fig. 21). The MR-Egger intercept provided only weak evidence of directional pleiotropy (intercept = 0.019, SE = 0.009, $P = 0.047$), and its small magnitude suggests any bias is likely minimal. Although modest heterogeneity was detected (Cochran's Q = 35.48, $P = 0.035$), the agreement of the causal estimate across methods indicates that the observed heterogeneity did substantially affect causal inference.

### Polygenic risk score analysis for nevi

We built a nevus PRS (Polygenic Risk Score) using SBayesRC applied to a meta-analysis of 13 out of the 14 nevus GWASs (excluding QSkin II) (see "Methods"). We tested the nevus PRS on the left-out QSkin II data ($n = 6307$). The model including nevus PRS explained 5% ($R^2 = 0.05$) of the variation, and a beta coefficient per (standard deviation) SD of the nevus PRS was 0.19 (SE = 0.0.01, $P < 4.94e^{-324}$), substantially exceeding the performance of the previous clumping PRS[15] ($R^2 = 0.0036$, beta = 0.051, SE = 0.01). Further, we applied the same nevus PRS to the Kidskin Young Adult Myopia Study (KYAMS) cohort, and the nevus PRS model performance was closely similar to that in QSKIN II, where $R^2$

was 4% and beta was 0.20 (SE = 0.060, $P$ = 0.0009, $n$ = 265). Ocular nevi are a risk factor for the rare and deadly uveal melanoma. We tested whether our PRS for cutaneous nevi was associated with iris nevi scored in the QIMR Brisbane Twin Nevus Study (BTNS) cohort. After generating a new PRS without the BTNS cohort, this PRS was weak but significantly associated with iris nevus count ($R^2$ = 0.0018, beta per SD of the PRS = 0.03, SE = 0.016, $P$ = 0.048, $n$ = 2607).

## Discussion

In this study, we performed the largest nevus GWAS meta-analysis to date identifying 29 genome-wide significant genomic loci. Of these, 24 had not previously been reported in a prior GWAS conducted solely for nevus count[5]. Prior analyses have leveraged the strong genetic overlap between melanoma and nevus count to test for regions of the genome pleiotropically associated with both traits. Promisingly, most of the loci newly identified for nevus count alone (19/24) were significant in these combined analyses[5,6]. Of the remaining five loci, two (TPCN2 and MAP3K1) had been identified in a joint analysis of melanoma and hair color[6] while the other three loci (BCAS2/SIKE1, AKAP13/KLHL25, and PTTG1) are completely novel and had not been identified in previous GWASs conducted for nevus count, melanoma, or pigmentation[6]. BCAS2, known to be amplified and overexpressed in breast cancer cell lines[16,17] may point to a potential role in cell proliferation that may contribute to nevus development. SIKE1 is involved in regulating immune responses, particularly those triggered by viruses and TLR3 (Toll-like receptor 3)[18]. SIKE1 may influence nevi growth by potentially affecting the immune-mediated control of melanocyte proliferation. AKAP13, identified as a tumor suppressor cooperating with PTEN in prostate cancer[19] may similarly influence nevus formation and melanoma risk. Finally, silencing of PTTG1 in melanoma cell lines impaired cell proliferation and invasiveness[20,21].

In addition, MAGMA gene-based analysis and TWAS identified four additional loci that were not genome-wide significant in the meta-analysis, including novel loci on chromosome 10 (LRMDA, also known as C10orf11) and chromosome 22 (C22orf26), for which the function is unclear. GDI2 was a strong candidate gene for one of the four additional loci with multiple evidence of association with nevi from TWAS and MAGMA, and prior evidence showed that a region near GDI2 (FAM208B) was significantly associated with a combined analysis of nevi and melanoma[6]. A melanocyte-differentiation gene, LRMDA when mutated, causes autosomal recessive albinism[22,23]. LRMDA is involved in melanocyte differentiation and melanosome maturation[23], which could provide insights into its genetic influence on nevus formation and possibly melanoma risk. MAGMA gene-based analysis identified NRAS as a significant candidate gene for the novel locus marked by rs7537232. Of note, NRAS codon 12/13 or 61 mutation is found in all (-15%) BRAF-mutant-negative acquired nevi[24]. NRAS is the second most common driver mutation in melanoma and is frequent in congenital melanocytic nevi[25].

As expected, when we performed our joint analyses of nevus count and melanoma, we found that the nevus-associated loci were pleiotropically associated with both traits; only one locus (TPCN2) did not strongly support any of the models. Based on the nevus count vs melanoma effect size plot for the identified 29 significant nevus loci (Supplementary Fig. 19), the MC1R variant (rs1805007) shows an inverse relationship; the allele that reduces nevus count increases the risk of melanoma. This may represent a true antagonistic biological effect on the two traits or may arise from the difficulty in identifying and counting nevi in individuals with very low skin pigmentation. As identified in a previous work, individuals within pale-skinned, highly freckled groups often show fewer nevi; however, molecular analyses have demonstrated that MC1R RHC (red hair color) homozygotes exhibit lower nevus counts as freckling increases across all freckling levels[26]. This pattern indicates that the association is unlikely to be explained solely by difficulty in counting nevi on lightly pigmented skin. The MC1R RHC variants likely reduce nevus formation by disrupting cAMP signaling, which is normally required for the early melanocyte proliferation that forms a mole[26] while increasing melanoma risk by impairing UV protection in melanocytes, making them more prone to malignant transformation. This opposing effect of MC1R variants, reducing nevus count yet increasing melanoma risk, may highlight a biological feature that warrants further investigation.

We also performed in-silico assays to determine likely candidate genes at identified loci. According to gene-set analysis in FUMA, prioritized genes (e.g., PLA2G6, CDKN2A, CCND1, GPRC5A, and SYNE2) from three gene mapping approaches (positional, eQTL, and chromatin interaction mapping) showed enrichment in known pigmentation-related traits such as hair color and tanning response (e.g., MAP3K1, PLK2, IRF4, EXOC2, DOCK8, TMEM38B, and TPCN2), reinforcing the shared genetics of nevus development and melanoma[5,6,27,28]. As well as known pathways, prioritized genes showed enrichments in several immune response-related pathways (e.g., the biological process involved in regulating peptidyl-serine phosphorylation of STAT protein, Supplementary Data 7) that may be critical in nevus formation and progression to melanoma, as immune dysregulation can drive tumor development through altered cell growth and inflammation.

Further, we identified gene set enrichments in cancers that do not have a pigmentation component. This finding underscores the significance of analyzing nevus count, as it can reveal biological pathways relevant to melanoma that fall outside the known risk pathways (i.e., skin color). In addition, MAGMA gene-set analysis, as well as FUMA gene-set analysis, revealed that candidate genes are significantly overrepresented especially in brain tumors, particularly glioblastomas and astrocytomas with neurofibromatosis type 1 (Supplementary Data 6 and 7). A connection between melanoma and these brain tumors has been observed in families with multiple melanomas and atypical mole syndrome[29]. Since melanocytes originate from the neural crest[30], they share a developmental lineage with certain brain cells, which may explain this association. In addition, there is an association between nevi and neurofibromatosis type 1[31]. These facts show that studying nevi can reveal biological pathways intersecting with brain function and neurological diseases. This shared origin positions nevi as a valuable focus for exploring connections between melanoma, brain biology, and related disorders.

In addition, the biological pathway associated with responding to organic compounds suggests that the genes enriched in this set likely play a significant role in responding to DNA damage, particularly from ultraviolet (UV) exposure. For instance, AHR and CYP1B1 were included in this gene set. AHR plays a role in melanoma risk through its interaction with UV exposure[32]. CYP1B1, an AHR target, has a risk variant that impacts melanoma susceptibility[32] and has been found to interact with environmental exposures (such as cook stove smoke) linked to increased cancer risk[33]. Accordingly, the candidate genes that are enriched in this pathway may offer potential target therapeutics that enhance DNA repair or mitigate UV-induced damage.

Sex differences in nevus count vary between body sites[7]. In our study, we observed a higher total body nevus count in females than in males. Despite this difference in count, the sex-stratified GWAS identified only two loci with heterogeneous effects. Further, the genetic correlation of nevus count in males and females was not significantly different from 1, indicating a largely shared genetic architecture. Thus, the observed sex difference may not be driven by sex-specific common genetic variants but rather reflect environmental or behavioral influences.

With minimal evidence of pleiotropy, MR analysis confirmed that genetic liability to higher nevus count contributes causally to melanoma risk, consistent with the previous observational and genetic evidence[15,34]. This shared common genetic basis also aligns with and reinforces our pairwise GWAS results.

We constructed a PRS for nevi that showed promising predictive power in two independent adulthood cohorts: the older adulthood cohort QSkin II ($R^2 = 0.05$) and the young adulthood cohort KYAMS ($R^2 = 0.04$), indicating its effectiveness in capturing genetic variance associated with nevus development.

Iris nevi and atypical cutaneous nevi are risk factors for uveal melanoma, a rare but deadly form of melanoma in the eye. We thus explored whether a genetic predisposition to cutaneous nevi formation also contributes to the development of iris nevi. The cutaneous nevus PRS yielded a lower (but statistically significant) predictive power ($R^2 = 0.0018$) for iris nevi compared to what we observed for cutaneous nevi. This may be because the overlap in genetic factors influencing iris nevi is incomplete. Future research should explore these differences further and investigate whether modifications to the nevus PRS can enhance its predictive ability for iris nevi and eventually uveal melanoma.

The strengths and limitations of our study should be considered in interpreting the findings. Our study was limited to individuals of European ancestry; hence, it may not be readily extendable to other populations. Our analysis was limited to individuals who were melanoma-free at the time of the last follow-up. While we cannot rule out that some may have subsequently been diagnosed with melanoma, the numbers would be low and unlikely to bias our results. Our study presents notable strengths. First, the large sample size compared to the previous nevus GWASs provided robust statistical power to identify several novel loci associated with nevus count. Second, through applying several post-GWAS analyses, we identified candidate genes in tissues, melanocytes, and skin that are relevant to the nevus and melanoma biology. Third, owing to the improvement in statistical power, the out-of-sample prediction ability of our nevus PRS is promising. This demonstrates the potential utility of this nevus PRS in identifying genetic predispositions that span both cutaneous and eye nevi, which may enhance the risk assessment of deadly cutaneous melanoma and uveal melanoma.

In summary, our GWAS, the largest to date, identified novel loci solely associated with nevus count as well as biological pathways unrelated to pigmentation, which is promising for future research towards therapeutic targets for melanoma. Through our study, we constructed a nevus PRS with promising predictive ability and in the future, by integrating the nevus PRS into melanoma risk prediction models, we anticipate improved accuracy in melanoma risk assessment, leveraging shared genetic factors between nevus count and melanoma susceptibility for more precise risk stratification.

## Methods

### GWAS cohorts

We used GWAS summary statistics for nevus count from 11 studies that were reported in a previously published nevus GWAS meta-analysis of 52,236 individuals of European ancestry[5]. These studies were combined with new data from the QSkin Sun and Health Study cohort[35] Phase I (QSkin I, $n = 15,346$) and Phase II (QSkin II, $n = 6608$), and the Australian Genetics of Depression Study (AGDS, $n = 11,775$)[36]. A subset of QSkin I ($n = 12,930$) with nevus count that was non-overlapping and unrelated (identical-by-descent score <0.15) to the QSkin melanoma case-control set was included in a previous pleiotropic GWAS analysis with melanoma[6]. However, in the current GWAS of QSkin I, we only excluded QSkin melanoma cases, increasing the number of available samples for this analysis of nevus loci ($n = 15,346$). Details of each cohort, including genotyping, quality control, imputation, and phenotypic descriptions, are provided in Supplementary Methods, sections 1.1 and 1.2. All study participants over 18 years provided written informed consent; for those under 18 years consent was obtained from parents, and full details of ethics approvals (QSkin I: P1309 and P2034, QSkin II: P3434, AGDS: P2118) have been reported elsewhere[5,35,36]. Our analysis only included individuals of European ancestry with no melanoma history, similar to previously published nevus GWAS meta-analysis[5].

### Polygenic risk score application cohorts

**KYAMS.** The KYAMS study was a follow-up of participants (aged 25–30 years) in the Kidskin Study, which began in 2015 to determine the impact of sun exposure intervention during childhood on myopia and eye health in adulthood[37]. The Kidskin study was initiated in 1995, a non-randomized controlled trial designed to reduce sun exposure in children through an educational intervention for children aged 5 to 6 years from 33 schools in Western Australia[38]. The KYAMS was approved by the Human Research Ethics Committees of the University of Western Australia (RA/4/1/6807), and written informed consent was obtained from all the participants[37]. For our analysis, we used the number of melanocytic nevi on the right arm counted by a trained observer. Bio samples for 113 participants were analyzed in 2016, while 183 were analyzed in 2021 using the Infinium Global Screening Array. For quality control of the genotyped data, SNPs with a call rate <0.95, HWE $P < 10^{-6}$, and MAF (minor allele frequency) <0.01 were excluded. Population outliers were identified using PCA (principal component analysis) based on participants of known European ancestry, using the 1000 Human Genome Project reference panel. Samples that were more than six standard deviations from the PC1 and PC2 centroids were excluded. One from each pair of individuals with an identity-by-descent score >0.2 was removed. Post-quality control, additional genetic variants were imputed using the Trans-Omics for Precision Medicine (TOPMed) reference panel[39]. For our analysis, high-quality imputed data ($r^2 > 0.5$ and MAF > 0.01) and nevus count data were available for 265 individuals.

**BTNS iris nevi.** The BTNS cohort has measurements for iris nevi in addition to the skin nevi used in the nevus GWAS meta-analysis. The cohort comprises a sample of predominantly adolescent twins (age 9–23 years) and their parents from Southeastern Queensland, Australia. The cohort was initially established to investigate the genetic factors influencing nevus development and other risk factors for melanoma[40,41]. The study was approved by the QIMR Human Research Ethics Committee, and written informed consent was obtained from all participants. Genotyping was performed on Illumina 610-Quad arrays for 2327 individuals phenotyped to the end of 2007 and on Illumina CoreExome arrays for 934 individuals phenotyped from 2008 to May 2013. After applying a series of quality controls, genotyped data were imputed using 1000 Genomes version 3 as described elsewhere[5]. The frequency of iris nevi derived from high-resolution digital photos of the iris was evaluated on an ordinal scale; (1) an absence of iris nevi, (2) at least one iris nevus, (3) at least three iris nevi, (4) at least five iris nevi, based on the number of melanin accumulations on the anterior border layer of the iris[42]. Although both the right and left iris were photographed, only the score from the right iris was included in the analysis due to the high polychoric correlation (0.91 to 0.95) between the two sides as reported in a previous study[42]. Post-quality-controlled ($r^2 > 0.5$ and MAF > 0.01) imputed dosage data and iris nevus count data were available for 2607 individuals.

### GWAS for QSkin I, QSkin II, and AGDS

QSkin has two self-reported nevus measurements recorded as (1) a count on the left upper arm and (2) a 4-point categorical scale ("none", "few", "some", "many"). The categorical nevus variable can be analyzed as a linear or ordinal trait. We used QSkin I data to compare the performance of a linear model in the Scalable and Accurate Implementation of the generalized mixed model (SAIGE)[43] and the proportional odds logistic mixed model (POLMM) which is implemented for ordinal trait analysis[44]. Both methods control sample relatedness. The performance of these GWAS models was assessed by comparing the number of significant hits and the total SNP heritability

estimated using linkage disequilibrium score regression (LDSC)[45]. A linear model in SAIGE was selected (Supplementary Methods, section 1.2 and Supplementary Figs. 1 and 2) and was then used to conduct GWAS in the QSkin II (Supplementary Fig. 4) and AGDS cohorts (Supplementary Fig. 5).

### Heritability and LD score intercept estimation
We estimated the proportion of variance explained by all SNPs (SNP-based heritability) for nevus count using QSkin I through genome-based restricted maximum likelihood (REML) implemented in GCTA v.1.26[46]. Following quality control, a genetic relationship matrix (GRM) was constructed using genotyped SNPs with MAF > 0.01. The GRM was fitted in with a REML analysis with 10 PCs included as covariates; a GRM cutoff of 0.05 was then applied to exclude related people.

LDSC regression (v1.0.0) was conducted for nevus GWAS summary results using pre-calculated LD scores from a 1000 Genomes European reference population, to estimate the genomic inflation factor, and to estimate the SNP-based heritability for GWAS where only summary statistics were available[45].

### GWAS meta-analysis
Our meta-analysis was based on 14 GWASs of nevus count, with X chromosome data available in 8 of these GWASs (Supplementary Table 2 and Supplementary Methods, section 1.2). As discussed in the cohort details (Supplementary Methods, section 1.2), there were slight differences in the assessment of nevus count, individual-level genotyping, imputation and quality control criteria, and the statistical models and analytical software employed in each GWAS. Therefore, we conducted a preliminary analysis to demonstrate that each study was consistent for conducting a GWAS meta-analysis for the nevus count trait. Specifically, after aligning effect alleles across all studies, we performed inverse variance weighted Mendelian Randomization (IVW MR)[47] as implemented in the "MendelianRandomisation" R package. The nevus count of each QSkin I, QSkin II, and AGDS study was obtained on a similar 4-point scale self-reported phenotypic assessment with moderate to high levels of reproducibility[48,49] and the effect sizes of genome-wide significant SNPs were consistent across these three GWASs (Supplementary Fig. 18). Hence, we used the GWAS meta-analysis conducted for QSkin I, QSkin II, and AGDS (Supplementary Fig. 6) as the reference study, and its top independent genome-wide significant SNPs as the instrumental variables to perform IVW MR. Based on the study-specific IVW MR plots (Supplementary Figs. 7 and 17), all individual studies from Duffy et al. (2018) were consistent with the three new studies.

Because the measurement of nevus count varies across studies, the meta-analysis of nevus GWAS was conducted using a fixed-effect sample size weighted Z-score method as implemented in METAL v.2020-05-05[50]. We applied genomic control correction to the input GWAS summary statistics. Meta-analyses were performed separately on autosomes and X chromosome. As no genome-wide significant associations were detected on the X chromosome, all downstream post-GWAS analyses were restricted to autosomal results. For post-GWAS analysis that required effect sizes and standard errors, for each SNP, we estimated the effect size and standard error using the Z-statistics, as these were the only outputs available[51].

Before conducting any post-GWAS analyses, the following quality control steps were applied to the meta-GWAS summary statistics. First, GWAS variants driven by a single study were dropped to ensure SNPs were associated in multiple studies. The DENTIST (Detecting Errors iN analyses of summary staTISTtistics) method (v.1.3.0.0) was used with default settings to obtain improved GWAS summary statistics by reducing heterogeneity between GWAS summary data and LD reference data[52]. In DENTIST analysis, the LD structure was obtained using a UK Biobank European reference panel comprising 5000 randomly selected and unrelated individuals. In this step, 8,783,913 variants were

tested by applying the DENTIST method, and 124,531 variants were filtered out.

Then, the FUMA platform v.1.6.0[53] was used to identify independent significant SNPs and significant loci using pre-calculated LD structure from the 1000 Genomes Phase 3 European reference population ($n = 503$). Independently significant SNPs were defined as genome-wide significant ($< 5 \times 10^{-8}$) SNPs that were independent at LD of $r^2 < 0.1$. Genomic loci associated with nevus count were defined as LD blocks of independent significant SNPs within a 1 Mb window. In addition, candidate SNPs that were defined as the SNPs having $r^2 \geq 0.1$ with one of the independent significant SNPs and MAF 0.01, were used for further annotations in FUMA. Forest plots of genome-wide significant lead SNPs were observed to check that the effect sizes were consistent across studies (Supplementary Fig. 18).

### GWAS-PW for nevus count and melanoma
GWAS-PW v.0.21[11] was used on nevus GWAS meta-analysis data and previously published melanoma GWAS[6] to determine whether the significant genetic loci from nevus GWAS meta-analysis were pleiotropic for nevus count and melanoma. For both nevus count and melanoma, SNPs were assigned to LD blocks using the recommended boundaries provided by GWAS-PW (https://bitbucket.org/nygcresearch/ldetect-data). GWAS-PW estimates posterior probability (PPA) for four different models: a locus (i) has an association with nevus count only; (ii) has an association with melanoma only; (iii) has a shared association with both melanoma and nevus count through the same genetic variant; (iv) is associated with both melanoma and nevus count through independent genetic variants. Any genome-wide significant locus with PPA > 0.8 for hypothesis (iii) was defined as a pleiotropic locus for melanoma and nevus count.

### Functional annotation, gene-based test, and gene-set analysis
Candidate SNPs were annotated for functional consequences on genes based on Ensembl genes [build 85] annotated using ANNOVAR in FUMA[53].

Then, positional, expression quantitative trait loci (eQTL) and chromatin interaction mapping approaches were conducted in FUMA to prioritize genes. First, annotated SNPs were mapped to genes, considering the physical position of the SNP inside a gene using a 10-kb window. Further, eQTL mapping was conducted within a 1 Mb window (cis-eQTL) to test whether GWAS SNPs are associated with the expression of the gene at a false discovery rate (FDR) $\leq 0.05$. eQTL data included were restricted to skin tissue from TwinsUK and Genotype-Tissues Expression (GTEx) v8 (sun-exposed, not sun-exposed skin types and fibroblasts). Moreover, chromatin interaction mapping was performed to test whether there was a significant (FDR $\leq 1 \times 10^{-6}$) chromatin interaction between nevus-associated regions and the promoter of nearby genes. Promoters were defined as 250 bp upstream and 500 bp downstream of the transcription site. Although skin-specific chromatin data were not available in FUMA, we included chromatin interaction data for all other 14 tissues (e.g., ovary, bladder, aorta, and lung) and seven cell types (e.g., embryonic stem cell, liver, mesenchymal stem, and fibroblast). This approach enabled a comprehensive gene mapping framework and the capture of genes overlapping with chromatin interaction regions that may be commonly expressed across tissues. Genes were prioritized only if supported by overlapping evidence from positional, eQTL, and chromatin interaction mapping.

In addition, MAGMA (Multi-marker Analysis of GenoMic Annotation)[54] gene-based and gene-set analyses were carried out as implemented in FUMA[53]. In the gene-based analyses, SNPs were mapped to 18,915 protein-coding genes obtained from Ensembl build 85. After accounting for LD, the association test statistics for SNPs were combined into a single gene-level $P$ value. Gene-wide significance was defined at $P = 0.05/18915 = 2.64 \times 10^{-6}$.

Here, we conducted two separate gene-set analyses using candidate genes selected from MAGMA gene-based analysis and three gene mapping approaches. In the MAGMA gene-set analysis, MAGMA gene-based $P$ values were further combined into each gene set from 4761 curated gene sets and 5917 gene ontology (go) terms from MsigDB v2023[53]. Significant gene sets were selected based on Bonferroni-corrected $P$ values. In addition to MAGMA gene-set analysis, genes prioritized by any of three gene mapping techniques were tested for overrepresentation in gene sets obtained from MsigDB, WikiPathways, and the GWAS catalog if at least two prioritized genes overlapped with a testing gene set[53]. $P$ values of gene sets were adjusted using the Benjamini–Hochberg correction.

## Transcriptome-wide association study and colocalisation analysis for nevus count

As an alternative approach to identifying candidate genes, a TWAS was performed for the nevus GWAS meta-analysis. eQTL data was sourced from skin tissue in GTex v8 (sun-exposed skin; $n = 508$; not-sun-exposed skin; $n = 430$, and fibroblasts; $n = 403$), whole-blood ($n = 31,684$) from eQTLGen[55] and a set of 106 primary melanocyte samples[6,56]. For melanocytes and other skin tissues in GTEx, pre-computed gene expression weights are available in a previous study[6,56] and FUSION website (http://gusevlab.org/projects/fusion/) respectively, and we performed TWAS for these tissues in FUSION using them. For whole-blood tissue, only eQTL summary-level data are available from the eQTLGen consortium. Since FUSION requires individual-level expression data to compute expression weights, we performed TWAS for whole-blood tissue in SUMMIT v.1.0.2 (Summary-level Unified Method for Modeling Integrated Transcriptome), which supports summary-level expression data[57]. This analysis utilized pre-calculated gene expression imputation models in SUMMIT, specifically built for eQTLGen data (https://doi.org/10.17605/OSF.IO/7MXSA).

For each set, a Bonferroni-corrected $P$ value threshold was defined as $0.05/($no of tested genes$)$ in all five tissue sets (melanocytes; 3797 genes, three skin tissues; $(9573 + 9556 + 10661) =$ genes, whole blood; 11415 genes, $P$ value is $0.05/(3797 + 9573 + 9556 + 10661 + 11415) = 1.11 \times 10^{-6}$).

Colocalisation analysis was conducted to estimate the posterior probability of a shared putative causal genetic variant between gene expression and nevus count (if PP4 $\geq 0.75$), as a complementary analysis for TWAS[58]. For melanocytes and skin tissues, colocalisation analyses were conducted within FUSION, which includes an interface to COLOC v.5.1.0.1 R package. A $P$ value threshold (--coloc_P flag) of 0.05 was used to compute COLOC statistics in FUSION. For eQTLGen data, colocalisation analysis was performed using COLOC v.5.1.0.1 package in R v.4.2.0 with default priors (p1 = $10^{-4}$, p2 = $10^{-4}$, p12 = $10^{-5}$; the prior probabilities that an SNP is significantly associated with the expression level of a gene, the trait outcome, or both, respectively).

## Nevus counts by sex

We tested the difference in nevus count across sex using data from QSkin I ($n = 15,346$, Supplementary Table 3). Nevus count data were on a 4-point scale; therefore, the ordinal logistic regression model was used to test the effect of sex, after adjusting for age and ancestry (top 10 PCs) in R v.4.3.1. Then, potential sex-specific genetic effects were assessed in autosomal SNPs and X chromosome using data from QSkin I, QSkin II, and AGDS. Sex-stratified GWASs were conducted for all three datasets using SAIGE (v.1.1.4) and then meta-analyzed using the inverse-variance weighted fixed-effect method in METAL v.2020-05-05[50]. To determine the statistical difference of sex-specific genome-wide significant loci in males and females, we performed a z-test[59] that compares the beta-estimates for the top lead SNPs in loci from sex-stratified meta-analyzed GWASs identified by FUMA v.1.6.0. Genetic correlation between males and females for nevus count was assessed using LDSC regression as described under "Heritability and LD score intercept estimation".

## Mendelian randomization for nevus count on melanoma risk

We used MR to assess the causal association between genetically determined nevus count and melanoma risk. Summary statistics for the outcome (melanoma) were obtained from a previous larger GWAS meta-analysis[6], and the analysis was restricted to a clinically-confirmed set including 30,134 cases and 81,415 controls of European ancestry. Genetic instruments were selected by clumping SNPs associated with nevus count ($P < 5 \times 10^{-8}$) from the current GWAS meta-analysis, based on LD ($r^2 < 0.001$ within a 10 mb window), using a reference panel from UK Biobank that included 5000 randomly selected unrelated individuals (37 SNPs). Further, we excluded ambiguous SNPs with allele frequency above 0.42. The F-statistic values > 10 were used to select strong instruments evaluating the relevance assumption. Cochran's Q statistic was used to assess heterogeneity in overall SNP effect estimates; if significant heterogeneity was detected, we excluded SNPs with SNP-level Q statistics exceeding 3.84[60]. We estimated the causal association between nevus count and melanoma using the inverse variance weighted method and performed sensitivity analyses using MR-Egger regression, weighted median, both simple and weighted mode estimators, to provide valid estimates under different assumptions[47,61]. Further, the presence of unbalanced horizontal pleiotropy was evaluated using the MR-Egger intercept. Analyses were conducted in R v.4.3.1 using the R package "TwoSampleMR".

## Polygenic risk score for nevus count

A PRS for nevus count was derived using SBayesRC, which uses functional genomic annotations to improve polygenic prediction[62]. To generate PRS for testing in young adults and older adults, as a discovery dataset, we conducted a nevus GWAS meta-analysis using 13 out of 14 cohorts, excluding QSkin II (final $n = 79,357$), whereas to predict iris nevus count, a separate (cutaneous) nevus GWAS meta-analysis was conducted, excluding BTNS (final $n = 82,704$) as above. In addition, LD data calculated from UK Biobank European ancestry and functional annotation data provided by SBayesRC (https://github.com/zhilizheng/SBayesRC) were used in the SBayesRC analysis.

We used QSkin II with individuals aged above 35 years ($n = 6307$), KYAMS ($n = 265$) and BTNS ($n = 2,607$) cohorts to determine the ability of nevus PRS to predict cutaneous nevus count in older adults (average age = 64 years, SD = 11 years, range = 35–93 years), young adults (average age = 28 years, SD = 1 year, range = 25–30 years), and then iris nevus count respectively. In addition, we used QSKIN II data to compare the prediction performance of the new SBayesRC-derived PRS with an LD-clumping-based PRS (collinearity $r^2 < 0.05$ and 2 Mb window) generated previously from the prior meta-analysis GWAS[5,15]. To score individuals in independent cohorts, imputed allelic dosages were weighted by the SBayesRC-derived SNP weights using PLINK v.2.0 (www.cog-genomics.org/plink/2.0/). We then standardized the nevus PRS to have a mean of zero and an SD of 1. Since related individuals were retained in all cohorts, GCTA-GREML was then used to regress inverse-normal rank-transformed cutaneous nevus count/iris nevus count, adjusted for sex, age, age$^2$, sex × age, sex × age$^2$, the first 10 PCs, and a GRM, onto standardised PRS. For Kidskin and BTNS cohorts, we used a binary batch variable as an additional covariate to control the genotype batch effect. Finally, nevus PRS performance was determined using $R^2$ computed by applying a formula reported in ref. 63.

## Reporting summary

Further information on research design is available in the Nature Portfolio Reporting Summary linked to this article.

## Data availability

Summary data for GWAS meta-analyses, including the sex-stratified meta-analyses, and nevus count PRS SNPs with weights, are available on Zenodo [https://doi.org/10.5281/zenodo.18537530][64]. All the individual GWAS summary statistics will be available from the corresponding authors upon request, subject to regulatory approval from the relevant institution. Individual-level genotype and phenotype data used in the heritability analysis and PRS analysis are protected and not publicly available due to data privacy laws. The UK Biobank reference panel was accessed under application number 25331. Access to UK Biobank data can be obtained by applying to UK Biobank (https://www.ukbiobank.ac.uk/). The dataset of LD blocks used in GWAS-PW analysis is available from https://bitbucket.org/nygcresearch/ldetect-data. LD score reference data can be accessed through https://alkesgroup.broadinstitute.org/LDSCORE/eur_w_ld_chr.tar.bz2. FUMA provides the reference panels and datasets used in the described analysis; identifying loci, functional annotation, gene-based and gene-set analysis through https://fuma.ctglab.nl/. Pre-computed gene expression weights for GTEx skin tissues are available on the FUSION website (http://gusevlab.org/projects/fusion/). Pre-calculated gene expression models for eQTL whole-blood data can be accessed through https://doi.org/10.17605/OSF.IO/7MXSA. SBayesRC provides LD data calculated from UK Biobank European ancestry and functional annotation data through https://github.com/zhilizheng/SBayesRC. GWAS for melanoma confirmed meta-analysis (with the exclusion of self-reported data from 23andMe and UK Biobank) is publicly available at dbGaP (phs001868.v1.p1).

## Code availability

All the data analyses were conducted using publicly available software programs and packages as detailed in "Methods".

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

## Acknowledgements

S.M. is supported by an Investigator grant (2034568) from the Australian National Health and Medical Research Council (NHMRC). D.C.W. is supported by an NHMRC Investigator grant (APP2026567). M.M.I. is supported in part by the National Institute for Health and Care Research (NIHR) Leeds Biomedical Research Centre (BRC) (NIHR203331). The views expressed are those of the author(s) and not necessarily those of the NHS, the NIHR, or the Department of Health and Social Care. S.E.M. is supported by NHMRC grants APP1172917 and APP2025674. N.K.H. is supported by the NHMRC (APP1195581), Jan Brown, the Buck Off Melanoma community, and memorial donors honoring Nicola Laws. S.S.Y.L. is supported by a Western Australia Future Health Research and Innovation Emerging Leaders Fellowship. This work was conducted using the UK Biobank Resource (application number 25331). We acknowledge the contribution of the data from the published nevus meta-analysis in 2018[5] and the melanoma meta-analysis in 2020[6]. All studies are forever grateful for the invaluable contributions of all research participants, their families, research nurses, research assistants, and support staff, whose dedication made this work possible. Please see Supplementary Methods for more details of acknowledgments and financial support for contributing studies.

## Author contributions

M.H.L. conceived the study and designed the analysis. M.H.L. and S.M. jointly supervised the study. G.J.M.S.R.J. carried out the analysis. G.J.M.S.R.J. and M.H.L. wrote the first draft of the manuscript. G.Z., N.P., C.M.O., N.G.M., P.A.L., S.E.M., S.D.G., S.D.T., G.L., S.S.Y.L., T.N., M.K., L.M. P., G.W.M., N.K.H., J.M.P., D.J.H., J.H., A.W.H., M.F., D.T.B., K.M.B.,

V.B., D.A.M., M.M.I., D.C.W., and D.L.D. were involved in cohort data collection. All authors critically reviewed the manuscript.

## Competing interests

The authors declare no competing interests.

## Additional information

[1]Genetics and Skin Cancer, QIMR Berghofer Medical Research Institute, Brisbane, QLD, Australia. [2]School of Biomedical Sciences, Faculty of Health, Medicine and Behavioural Sciences, The University of Queensland, Brisbane, QLD, Australia. [3]Genetic Epidemiology, QIMR Berghofer Medical Research Institute, Brisbane, QLD, Australia. [4]Cancer Control Group, QIMR Berghofer Medical Research Institute, Brisbane, QLD, Australia. [5]School of Public Health, Faculty of Health, Medicine and Behavioural Sciences, The University of Queensland, Brisbane, QLD, Australia. [6]Dermatology Research Centre, Frazer Institute, The University of Queensland, Brisbane, QLD, Australia. [7]Psychiatric Genetics, QIMR Berghofer Medical Research Institute, Brisbane, QLD, Australia. [8]School of Biomedical Sciences, Faculty of Health, Queensland University of Technology, Brisbane, QLD, Australia. [9]School of Psychology, Faculty of Health, Medicine and Behavioural Sciences, The University of Queensland, Brisbane, QLD, Australia. [10]Statistical Genetics, QIMR Berghofer Medical Research Institute, Brisbane, QLD, Australia. [11]Centre for Ophthalmology and Visual Science (incorporating the Lions Eye Institute), The University of Western Australia, Nedlands, WA, Australia. [12]Centre for Eye Research Ireland, Technological University Dublin, Dublin, Ireland. [13]Centre for Eye Research Australia, East Melbourne, VIC, Australia. [14]Department of Dermatology, Erasmus MC Cancer Institute, Erasmus MC University Medical Centre Rotterdam, Rotterdam, The Netherlands. [15]Department of Pathology and Clinical Bioinformatics, Erasmus MC Cancer Institute, Erasmus MC University Medical Centre Rotterdam, Rotterdam, The Netherlands. [16]Institute for Molecular Bioscience, The University of Queensland, Brisbane, QLD, Australia. [17]Oncogenomics, QIMR Berghofer Medical Research Institute, Brisbane, QLD, Australia. [18]Department of Epidemiology, Harvard School of Public Health, Boston, MA, USA. [19]Department of Epidemiology, Richard M. Fairbanks School of Public Health, Indiana University, 1050 Wishard Boulevard, Indianapolis, IN, USA. [20]Melvin and Bren Simon Comprehensive Cancer Center, Indiana University, 1050 Wishard Boulevard, Indianapolis, IN, USA. [21]Menzies Institute for Medical Research, School of Medicine, University of Tasmania, Hobart, TAS, Australia. [22]Centre for Eye Research Australia, Royal Victorian Eye and Ear Hospital, East Melbourne, VIC, Australia. [23]Department of Twin Research and Genetic Epidemiology, King's College London, London, UK. [24]Leeds Institute for Data Analytics, University of Leeds, Leeds, UK. [25]Division of Cancer Epidemiology and Genetics, National Cancer Institute, Bethesda, MD, USA. [26]Department of Dermatology, Hemel Hempstead Hospital, West Hertfordshire NHS Trust, Hemel Hempstead, UK. [27]NIHR Leeds Biomedical Research Centre, Leeds Teaching Hospitals NHS Trust, Leeds, UK. ✉e-mail: shanika.jayasinghe@qimrb.edu.au; Matthew.Law@qimrb.edu.au

