## [Transparent Peer Review file · Nature Communications]

A large-scale genome-wide association meta-analysis for nevus count provides direct insights into the genetics of melanoma

Corresponding Author: Ms G J M Shanika R Jayasinghe

Version 0:

Reviewer comments:

Reviewer #1

(Remarks to the Author)

This is an important study - the largest to date of genetic variant associations with nevi. The authors have identified new loci of interest.

I have just a few minor comments:

Acronyms: FUMA is not defined.

PRS not defined until page 20 although used earlier.

I think for more general audience understanding, it would be helpful to be very specific about the SNPs used in the PRS and their weights. Maybe a supplementary table with the PRS snps and the weights used would make this easier to understand and for researchers to apply to other datasets in the future. Also, how do these weights from SBayesRC compare with the weights previously used in these author's other papers? Also how does the PRS prediction compares to prior PRS scores. What do the extra SNPs and new method of weighting add to the prediction?

Reviewer #2

(Remarks to the Author)

In their paper "A large-scale genome-wide association meta-analysis for nevus count provides direct insights into the genetics of melanoma", Jayasinghe et al. performed the largest GWAS meta-analysis to date of nevus count using previously published data, and adding two more cohorts, all corresponding to European genetic ancestry. They uncovered novel loci associated with nevus count, which have not been associated previously with other melanoma or pigmentation traits, and performed gene-based, as well as TWAS and pathway analyses to understand the biological and molecular processes involved, and their relationship with melanoma. Additionally, they created polygenic risk scores for nevus count and additionally for iris nevi, which are a risk factor for uveal melanoma. Overall, the authors provide important contributions to the field of pigmentation and melanoma genetics, using a broad range of computational approaches.

Major comments

1. The authors performed GWAS in two new cohorts to perform a meta-analysis with previously published data. I understand that until recently, chromosome X in GWAS has been often neglected and rarely included, therefore I do not expect the authors to re-analyze other cohorts' data for chromosome X. However, it is now becoming fairly common, and GWAS tools such as SAIGE allow the inclusion of chromosome X in the analysis. Therefore, the authors should be able to easily include the X chromosome in the two new cohorts they analyze in the present paper, which will strengthen the paper (even if no signals are identified), and will help pave the way for other future studies to include it.

2. The authors should report the age range of participants across cohorts used in the meta-analysis and the proportion of participants across the sexes (at least in the two new cohorts included). It would also be relevant to

assess if there are phenotypic (and if so, genetic) differences between the sexes, as it is known that there are differences in the incidence of melanoma between males and females.

3. In Supplementary Note 1.2, the authors argue that the SNP-based heritability estimates were slightly higher for their SAIGE approach, compared to POLMM, hence choosing the SAIGE approach for their downstream analyses. However, considering the SE of the estimates, they are basically the same. Therefore, the argument to choose SAIGE over POLMM seems a little weak. On one hand, one can argue that SAIGE is one of the standard GWAS mixed-models tools used by the community, whereas on the other hand, a model that considers an ordinal scale would be the better choice due to the nature of the data used for the GWAS.

4. The authors should provide more details in the Methods regarding their colocalization analyses, specifically the priors utilized. Additionally, in the Results section there is no mention of the follow-up results of TWAS with colocalization. The authors should specify in their TWAS results that candidate genes were selected based on TWAS followed-up by colocalization, if that is the case (which seems to be). Finally, from the main text Results, the authors should specify which TWAS analyses (i.e., which cell-type/tissue) yielded the significant results and if the direction of effect was concordant when the signal was in multiple TWAS analyses (i.e., GDI2), especially given the less relevant whole-blood dataset used.

5. The authors should provide the reasoning behind the inclusion of 14 different tissues for the chromatin interaction mapping, given that this is a broad list of tissues/cell types that do not necessarily have an evident relation to skin.

6. The finding of a possibly antagonistic effect between nevus count and melanoma in the MC1R locus is pretty interesting. As an alternative explanation, the authors mention in the Discussion that this might be a consequence of the difficulty in identifying nevi in individuals with red hair and very low skin pigmentation. However, I do not see how specifically red (scalp) hair can possibly influence nevi count (as the authors suggest). In contrast, I could imagine how the presence of freckles (where MC1R is involved too) could influence nevi count.

7. One of the main conclusions of the authors is the possibility to include nevus count genetic signals as risk factors for melanoma PRS, and they have shown with pair-wise analysis the overlap of the genetic signals between both traits. However, including a causal inference approach, such as Mendelian randomization, using the nevus count genetic signals as exposures and melanoma GWAS as outcome, would aid in understanding the relationship between these two traits, and would potentially strengthen the conclusions of the paper.

Minor comments

1. Citations in the introduction are in different format (i.e., First sentence of second paragraph).

2. In S1.2, there is a typo 'identify-by-descent' should be 'identity-by-descent'.

3. Readers would appreciate if authors could specify the genome build of the genomic coordinates used in the Supplementary Tables. Additionally, it would be appreciated if the column names throughout the Supplementary Tables could be better described, as I only see an 'Abbreviations' tab for one table.

4. The authors should re-organize the order in which Supplementary Tables appear. It is strange that the first Supplementary Table referenced in the Results section of the main text is Supplementary Table 7, whereas other Supplementary Tables are not referenced throughout the main text. Similarly, the Methods section for GWAS-PW should be put in the same order as in the Results section.

5. In the Functional Mapping Results section, there seems to be a word missing from the sentence: "Of 248 genes, 26 mapped on 14 of 29 nevus-associated genomic loci that were identified by positional, eQTL and chromatin interaction mapping".

6. For the nevus PRS, what were the age ranges selected to classify in older vs. young adults? The authors should specify the age ranges in the Methods, not only the average age of each cohort.

Version 1:

Reviewer comments:

Reviewer #1

(Remarks to the Author)

All questions have been satisfactorily answered. The work is significant to the field, the conclusions are supported. The methodology is sound and there is enough detail.

Reviewer #2

(Remarks to the Author)

The authors have made a significant improvement to their manuscript, and it reads very well.

I just have one minor comment left, but otherwise I recommend for publication, with regards to the use of gender terms in the context of sex-specific GWAS. If the authors refer to genetic sex, then the correct terms to use would be female/male instead of women/men.

Reviewer(s)' Comments to Author:

Referee: 1

Reviewer #1 (Remarks to the Author):

This is an important study - the largest to date of genetic variant associations with nevi. The authors have identified new loci of interest.

I have just a few minor comments:

1. *Acronyms: FUMA is not defined.*

Response: Corrected by defining FUMA at the first occurrence (Page 6, line 134)

2. *PRS is not defined until page 20, although used earlier.*

Response: Corrected by defining PRS at the first occurrence (Page 11, line 241)

3. *I think for a more general audience understanding, it would be helpful to be very specific about the SNPs used in the PRS and their weights. Maybe a supplementary table with the PRS SNPs and the weights used would make this easier to understand and for researchers to apply to other datasets in the future.*

Response: Together with the meta-analysis GWAS summary statistics, the PRS SNPs with weights will be available in Zenodo, as the PRS includes 7,280,843 SNPs. The data availability statement has been updated to more clearly report this.

4. *Also, how do these weights from SBayesRC compare with the weights previously used in these authors' other papers?*

Response: The weights used in the current PRS differ from those in the authors' previous publications for two main reasons. First, the meta-analysis methods and thus the weight scale differ: the previous PRS (e.g. used in PMID: 34778946) was derived from an inverse-variance weighted meta-analysis, whereas the current PRS was based on a p-value-based meta-analysis. Second, whereas the previous PRS used a limited number of independent genome-wide SNPs (8 SNPs), SBayesRC integrates over ~7 million SNPs and applies LD-aware modelling combined with functional annotation-informed Bayesian shrinkage to produce more accurate SNP effect estimates, and to improve polygenic prediction of traits. As a result, direct comparison of the weights is not straightforward. However, they are more readily compared using their predictive ability, which is answered in the reviewer's next question.

5. *Also how does the PRS prediction compares to prior PRS scores. What do the extra SNPs and new method of weighting add to the prediction?*

Response: Compared with the 2018 clumping-based PRS ($R^2 = 0.36\%$, $\beta = 0.051$, $SE = 0.01$), the SBayesRC-derived PRS shows substantially improved predictive performance ($R^2 = 5\%$, $\beta = 0.19$, $SE = 0.01$). The results were added to the manuscript under the PRS analysis (Page 11, lines 245-246).

Referee 2:

Reviewer #2 (Remarks to the Author):

In their paper “A large-scale genome-wide association meta-analysis for nevus count provides direct insights into the genetics of melanoma”, Jayasinghe et al. performed the largest GWAS meta-analysis to date of nevus count using previously published data, and adding two more cohorts, all corresponding to European genetic ancestry. They uncovered novel loci associated with nevus count, which have not been associated previously with other melanoma or pigmentation traits, and performed gene-based, as well as TWAS and pathway analyses to understand the biological and molecular processes involved, and their relationship with melanoma. Additionally, they created polygenic risk scores for nevus count and additionally for iris nevi, which are a risk factor for uveal melanoma. Overall, the authors provide important contributions to the field of pigmentation and melanoma genetics, using a broad range of computational approaches.

Major comments

1. The authors performed GWAS in two new cohorts to perform a meta-analysis with previously published data. I understand that until recently, chromosome X in GWAS has been often neglected and rarely included, therefore I do not expect the authors to re-analyze other cohorts’ data for chromosome X. However, it is now becoming fairly common, and GWAS tools such as SAIGE allow the inclusion of chromosome X in the analysis. Therefore, the authors should be able to easily include the X chromosome in the two new cohorts they analyze in the present paper, which will strengthen the paper (even if no signals are identified), and will help pave the way for other future studies to include it.

Response: We thank the reviewer for the helpful suggestion. We agree that the X chromosome has been under-analysed in GWAS, but that including it is increasingly feasible and will strengthen the field moving forward. We have now included an analysis of the X chromosome in QSkin I, QSkin II and AGDS using SAIGE. In addition, data on the X chromosome were available from 5 GWASs included in the previous meta-analysis (PMID: 30429480). We then integrated these GWASs into a meta-analysis of chromosome X. However, no genome-wide significant associations were detected in either cohort or the meta-analysis. We have reported the methods and results related to this analysis in the manuscript and will include chromosome X results in the shared GWAS summary statistics (Methods: Page 21, line 474 and Supplementary Note S1.2 Page 29; Results: Page 5, lines 103-106)

2. The authors should report the age range of participants across cohorts used in the meta-analysis and the proportion of participants across the sexes (at least in the two new cohorts included). It would also be relevant to assess if there are phenotypic (and if so, genetic) differences between the sexes, as it is known that there are differences in the incidence of melanoma between males and females.

Response: The age ranges were available for 11 studies from the previous meta-analysis (PMID: 30429480), and so were added to Table S1.1 in Supplementary Note (Page 25-26). Age ranges and proportions of participants across sex were reported for the new cohorts under the cohort description (Supplementary Note S1.2: Page 27-28).

We also thank you for your thoughtful comment regarding the addition of sex-specific analyses to the manuscript. We assessed the phenotypic differences in the nevus count across sex using QSkin I data and observed a higher total-body nevus count in women than in men. Therefore, we performed sex-specific GWAS by meta-analysing QSkin I, QSkin II, and AGDS to identify the difference in the genetic architecture of nevus count between men and women. The genetic correlation was not significantly different to 1 ($rg = 0.863$, 95% CI = 0.453 – 1.273), indicating that the genetic architecture of nevus count was largely shared across men and women. We have updated the methods section and added the results to the manuscript under “Sex-specific GWAS meta-analysis for nevus count” (Methods: Page 26, lines 570-582; Results: Page 10, lines 213-227; Discussion: Page 15, lines 333-338).

3. In Supplementary Note 1.2, the authors argue that the SNP-based heritability estimates were slightly higher for their SAIGE approach, compared to POLMM, hence choosing the SAIGE approach for their downstream analyses. However, considering the SE of the estimates, they are basically the same. Therefore, the argument to choose SAIGE over POLMM seems a little weak. On one hand, one can argue that SAIGE is one of the standard GWAS mixed-models tools used by the community, whereas on the other hand, a model that considers an ordinal scale would be the better choice due to the nature of the data used for the GWAS.

Response: We agree that the SNP-based heritability estimates (h^2) from SAIGE and POLMM are very similar when considering their standard errors. While the SEs overlapped, we acknowledge that selecting SAIGE based on the fact that its point estimate was higher has the advantage of aligning with the generally used method. We have revised the text in the supplementary note 1.2 to acknowledge these facts (Supplementary Note S1.2: Page 28).

4. *The authors should provide more details in the Methods regarding their colocalization analyses, specifically the priors utilized. Additionally, in the Results section there is no mention of the follow-up results of TWAS with colocalization. The authors should specify in their TWAS results that candidate genes were selected based on TWAS followed-up by colocalization, if that is the case (which seems to be). Finally, from the main text Results, the authors should specify which TWAS analyses (i.e., which cell-type/ tissue) yielded the significant results and if the direction of effect was concordant when the signal was in multiple TWAS analyses (i.e., GDI2), especially given the less relevant whole-blood dataset used.*

Response: We have expanded the Methods section to detail the priors used in our colocalization analyses (Page 25-26, lines 566-568). In the Results, we specified that candidate genes were identified through TWAS, followed by colocalization, along with the tissues that showed significant TWAS associations for each gene. For genes like GDI2, which exhibited signals in multiple tissues, we have included the directions of effect and noted their concordance across tissues in the results (Page 8-9, lines 184-190).

5. *The authors should provide the reasoning behind the inclusion of 14 different tissues for the chromatin interaction mapping, given that this is a broad list of tissues/ cell types that do not necessarily have an evident relation to skin.*

Response: Chromatin interaction mapping was performed using FUMA. Since skin-specific chromatin data were not available in FUMA, we included all other available 14 tissues and 7 cell types to provide a comprehensive gene-mapping framework and to capture genes that overlapped with chromatin interaction regions that may be commonly expressed across tissues. Importantly, genes were prioritised only if supported by overlapping evidence from chromatin interaction, positional mapping, and eQTL mapping conducted using skin tissues. We have added this clarification to the methods (Page 24, lines 522-528).

6. *The finding of a possibly antagonistic effect between nevus count and melanoma in the MC1R locus is pretty interesting. As an alternative explanation, the authors mention in the Discussion that this might be a consequence of the difficulty in identifying nevi in individuals with red hair and very low skin pigmentation. However, I do not see how specifically red (scalp) hair can possibly influence nevi count (as the authors suggest). In contrast, I could imagine how the presence of freckles (where MC1R is involved too) could influence nevi count.*

Response: We thank the reviewer for this thoughtful comment. We agree that attributing reduced nevus counts specifically to red hair needs to be correct; if there is a role, it is likely to act on the skin or moles directly. We appreciate the suggestion to also consider the role of freckling more directly. As noted in a previous work, individuals within pale-skinned, highly freckled groups often show fewer nevi; however, molecular analyses have demonstrated that *MC1R* RHC homozygotes exhibit lower nevus counts as freckling increases across all freckling levels. This pattern indicates that the association is unlikely to be explained solely by difficulty in counting nevi on lightly pigmented skin. The *MC1R* RHC variants likely reduce nevus formation by disrupting cAMP signalling, which is normally required for the early melanocyte proliferation that forms a mole (PMID: 14709592), while simultaneously increasing melanoma risk by impairing UV protection in melanocytes. We have revised the Discussion to reflect this mechanistic, rather than phenotypic-artifact, explanation (Page 13, lines 289-300).

7. *One of the main conclusions of the authors is the possibility to include nevus count genetic signals as risk factors for melanoma PRS, and they have shown with pair-wise analysis the overlap of the genetic signals between both traits. However, including a causal inference approach, such as Mendelian randomization, using the nevus count genetic signals as exposures and melanoma GWAS as outcome, would aid in understanding the relationship between these two traits, and would potentially strengthen the conclusions of the paper.*

Response: We thank the reviewer for this helpful suggestion. We have now performed Mendelian randomisation using nevus count–associated variants as instruments and used a previous melanoma GWAS (PMID: 32341527) as the outcome. This found that genetically predicted higher nevus count was significantly associated with a higher risk of melanoma (Inverse variance weighted (IVW) OR = 4.212, 95% CI = 3.469-5.114). We have updated the results and methods sections, integrating MR analysis into the manuscript (Methods: Page 26-27, lines 584-600; Results: Page 10-11, lines 229-238; Discussion: Page 15, 339-342).

Minor comments

8. Citations in the introduction are in different format (i.e., First sentence of second paragraph).

Response: Corrected (Page 3, line 71).

9. In S1.2, there is a typo 'identify-by-descent' should be 'identity-by-descent'.

Response: Corrected.

10. Readers would appreciate if authors could specify the genome build of the genomic coordinates used in the Supplementary Tables. Additionally, it would be appreciated if the column names throughout the Supplementary Tables could be better described, as I only see an 'Abbreviations' tab for one table.

Response: Genome build has been added to each table. We have also added readable column names.

11. The authors should re-organize the order in which Supplementary Tables appear. It is strange that the first Supplementary Table referenced in the Results section of the main text is Supplementary Table 7, whereas other Supplementary Tables are not referenced throughout the main text. Similarly, the Methods section for GWAS-PW should be put in the same order as in the Results section.

Response: We have reorganised the Supplementary tables and revised the methods section accordingly. The order of the Methods section for GWAS-PW has been changed to align with the order of the Results section.

12. In the Functional Mapping Results section, there seems to be a word missing from the sentence: "Of 248 genes, 26 mapped on 14 of 29 nevus-associated genomic loci that were identified by positional, eQTL and chromatin interaction mapping".

Response: Corrected as "Of 248 genes, 26 overlapped in all three mapping approaches, and those genes mapped on 14 of 29 nevus-associated genomic loci". (Page 6 lines 139-141)

13. For the nevus PRS, what were the age ranges selected to classify in older vs. young adults? The authors should specify the age ranges in the Methods, not only the average age of each cohort.

Response: Thank you for this suggestion. Initially, we classified the two adult groups (younger vs. older) based on the cohort-level average ages: KYAMS (mean age = 28 years, SD = 1 year, range = 25–30 years) and QSKIN II (mean age = 63 years, SD = 13 years, range = 18–93 years). To provide a clearer separation between the two groups, we have now restricted the PRS analysis in the QSKIN II cohort to individuals aged >35 years. We have specified the age ranges for both cohorts in the Methods section (Page 27, lines 613-614).

Reviewer(s)' Comments to Author:

Referee: 1

Reviewer #1 (Remarks to the Author):

All questions have been satisfactorily answered. The work is significant to the field, the conclusions are supported. The methodology is sound and there is enough detail.

Response: We thank the reviewer for the positive assessment and glad that all questions have been satisfactorily addressed.

Additionally, we would like to note that we have improved the clarity of the data access statement in general without changing the access status.

Referee 2:

Reviewer #2 (Remarks to the Author):

The authors have made a significant improvement to their manuscript, and it reads very well.

Response: We thank the reviewer for the positive feedback.

I just have one minor comment left, but otherwise I recommend for publication, with regards to the use of gender terms in the context of sex-specific GWAS. If the authors refer to genetic sex, then the correct terms to use would be female/male instead of women/men.

Response: We have updated the manuscript to use female and male when referring to genetic sex, replacing women and men where applicable.

Additionally, we would like to note that we have improved the clarity of the data access statement in general without changing the access status.